

# The relationships between bilingual learning, willingness to study abroad and convergent creativity

Yuan Zhao[1,*], Yuan Yuan[2], Wangbing Shen[3], Chuanlin Zhu[1] and Dianzhi Liu[1,*]

[1] School of Education, Soochow University, Suzhou, China
[2] School of Rehabilitation Science, Jiangsu Provincial Key Laboratory of Special Children's Impairment and Intervention, Nanjing Normal University of Special Education, Nanjing, China
[3] School of Public Administration and Business School, Hohai University, Nanjing, China
[*] These authors contributed equally to this work.

## ABSTRACT

Convergent creativity is a form of creative thinking that uses existing knowledge or traditional methods to analyze available information and generate an appropriate solution. The differences in the performance of participants in convergent creativity caused by bilingual learning is a popular research area in creativity. A final sample of 68 participants was asked to complete the remote associates test (RAT). The results indicate that a moderate positive correlation exists between bilingual learning and convergent creativity. Students who want to study abroad perform better on the RAT than those who do not, and this effect is mediated by second language proficiency. These findings suggest that improving students' English proficiency and increasing their opportunities to study abroad may be effective ways to promoting convergent creativity.

# INTRODUCTION

## What is creativity?

Creativity is defined as the ability of a group or an individual to generate original and appropriate problem solutions (*Runco, 2007*; *Moraru, Memmert & Van der Kamp, 2016*). Creativity is an advanced expression of human intelligence and plays an important role in scientific creation, social progress and technological innovation. *Guilford (1967)*, a past president of the American Psychological Association, noted that creativity mainly includes divergent creativity and convergent creativity. The former refers to how individuals generate new information and produce a wide variety of outputs from the same source in a novel way. Divergent creativity is mainly evaluated by tasks such as the Torrance Test of Creative Thinking (TTCT) (e.g., *Humble, Dixon & Mpofu, 2018*) and the Alternative Uses Test (AUT) (e.g., *Hao et al., 2017*). In contrast, convergent creativity refers to the cognitive process in which individuals use existing knowledge or traditional methods to analyze given information and obtain the best answer (*Lee & Therriault, 2013*; *Ritter, Abbing & Van Schie, 2018*). The processes of convergent creativity include the evaluation of the divergent creativity stage and the selection of ideas, or a full range of phases ranging from

Corresponding authors
Yuan Yuan, psychyy1989@163.com
Wangbing Shen, wangbingsh09@163.com

evaluation to implementation (*Coursey et al., 2018*). In real life, many different tasks rely on the use of convergent creativity because this type of creativity is part of creation, and the skillful use of convergent creativity can be critical for creative idea production (*Vries & Lubart, 2019*). The emphasis on convergent creativity is largely attributed to Simon's study investigating problem solving and the application of Mednick's study on the RAT (*Shen et al., 2015*). Currently, convergent creativity is primarily measured by tasks such as classical insight problems (e.g., *Chein et al., 2010*) and the RAT (e.g., *Alici et al., 2019*; *Shen et al., 2016a*). The RAT was used as the experimental material in this study, which originated from the theoretical model proposed by Mednick in 1962 about the relationship between associative behavior and creativity (*Benedek & Neubauer, 2013*). Guilford highlighted the importance of divergent creativity, which has led convergent creativity to be overlooked. Although convergent creativity has long been on the outskirts of creative research, with the rapid development of research on brain-based insight, an increasing number of scholars have begun to pay attention to this important issue (*Shen et al., 2015*).

## The influence of bilingual learning and willingness to study abroad on creativity

Accompanying the development of globalization and international trade, cultural migration and exchange activities such as immigration, transnational study, and multilanguage learning have become increasingly frequent (*Cheng & Leung, 2013*; *Shen & Yuan, 2015*). In different countries, people need to understand and absorb as well as accept the influence of different cultures, which might be an important requirement for people to adapt to the rapid development of modern society. Thus, cultural activities have an impact on people's ways of thinking.

Rohan, a young entrepreneur mentioned in *Tadmor, Galinsky & Maddux*'s study (*2012*), lived in five different countries. Rohan stumbled upon some cocoa plantations in Mexico and became fascinated by them. Subsequently, in France, Rohan discovered a chocolate shop that inspired his imagination. Using Aztec-themed designs, Rohan organically combined the pre-Hispanic roots of cacao and the chocolate products that we currently consume. This example highlights how multicultural experiences and exposure to new cultures inspire new ideas. Multicultural experiences refer to all direct and indirect experiences of contact or interaction with elements and/or members of foreign cultures (*Hu et al., 2017*); such experiences can facilitate the performance of individual creativity by improving individual enlightenment learning, distance imagination and concept formation and by generating novel ideas through unfamiliar culture (*Wang & Wang, 2018*). Under these circumstances, individuals draw new ideas from different cultures and integrate them in novel ways in problem-solving scenarios. Integrating seemingly unrelated concepts in different cultures helps to extend the conceptual category in the brain (*Maddux, Adam & Galinsky, 2010*; *Wang & Wang, 2018*). *Shen & Yuan (2015)* summarized Maddux's and other scholars' studies (*De Bloom et al., 2014*; *Fee & Gray, 2012*; *Maddux & Galinsky, 2009*; *Yi et al., 2013*) and concluded that both long-term and short-term multicultural experiences can boost individual creativity and that improvement in cognitive fluency is among the most obvious outcomes. There are two main reasons for this phenomenon: on the one hand, people

with both long- and short-term multicultural experiences have a wide understanding of different cultures, so they can better establish connections between ideas from different cultural sources. On the other hand, extensive exposure to different cultures can sometimes produce conflicting views. This state of mind helps multicultural individuals overcome fixed cognition, eliminate the structured and conventional methods of addressing problems, and stimulate creative thinking (*Cheng & Leung, 2013*). In addition, when individuals are exposed to foreign cultures with a learning mindset, they might elicit a comprehensive emotional response, such as the recognition of some excellent qualities or admiration for some achievements in foreign cultures, which can also enhance creativity (*Cheng, Leung & Wu, 2011*). In conclusion, in this era of globalization, multicultural experiences will create important cognitive benefits (*Cheng, Leung & Wu, 2011*; *Wang & Wang, 2018*).

Existing studies have shown that the length of time people have lived abroad and their overseas experiences can predict their creative ability (e.g., *De Bloom et al., 2014*; *Fee & Gray, 2012*; *Maddux & Galinsky, 2009*; *Shen & Yuan, 2015*). However, some studies have found that a certain proportion of those who travel abroad fail to achieve success in the new culture (e.g., *Mendenhall & Oddou, 1985*; *Wederspahn, 1992*). While some scholars believe these failures may be due to the lack of adjustment or the psychological discomfort experienced when living abroad (*Maddux & Galinsky, 2009*), evidence regarding the relationship between adjustment and performance is contradictory (*Tadmor, Galinsky & Maddux, 2012*). One potential reason is the willingness to study or stay abroad. Some individuals might have no willingness to study or stay abroad (a lack of openness to new countries, cultures, or things) even if they are in foreign countries. Undoubtedly, many people—including students, teachers, and technicians—plan to study abroad but are still in the process of preparation. As Fig. 1[1] reports with respect to Chinese students studying abroad, as Chinese society develops, an increasing number of people have taken advantage of the opportunity to study abroad, and people increasingly study or plan to study abroad (*Li & Sun, 2018*). The intention to study abroad may be related to creativity, and students who have plans to study abroad usually have better creative performance than those who do not have such plans (*Lee, Therriault & Linderholm, 2012*). However, few studies to date have explored this interesting issue. Based on existing studies, we are inclined to support the results of *Lee, Therriault & Linderholm (2012)*.

The main reason for the hypothesis that the intention to study abroad may be related to creativity is that Chinese students who want to study or work abroad must improve their English language proficiency. Many studies have confirmed that bilingual learning can promote creativity (e.g., *Cummins, 1979*; *Hommel et al., 2011*; *Wang & Cheng, 2016*). From the perspective of students' creativity in English learning, English proficiency is the most important factor determining students' ability to create in English (*Wang & Cheng, 2016*). In fact, many studies have confirmed the influence of second language proficiency on creativity in children. *Kharkhurin (2011)* used thirteen examples from 1966 to 1999 to illustrate that bilingual children outperformed monolingual children in divergent thinking traits such as fluency, flexibility, elaboration, and originality. Similarly, the effect of second language proficiency on creativity has been supported by some studies involving adults. For example, the performance of bilingual college students on the Abbreviated Torrance Test of

[1] Data were collected from the National Bureau of Statistics of China (http://www.stats.gov.cn/).

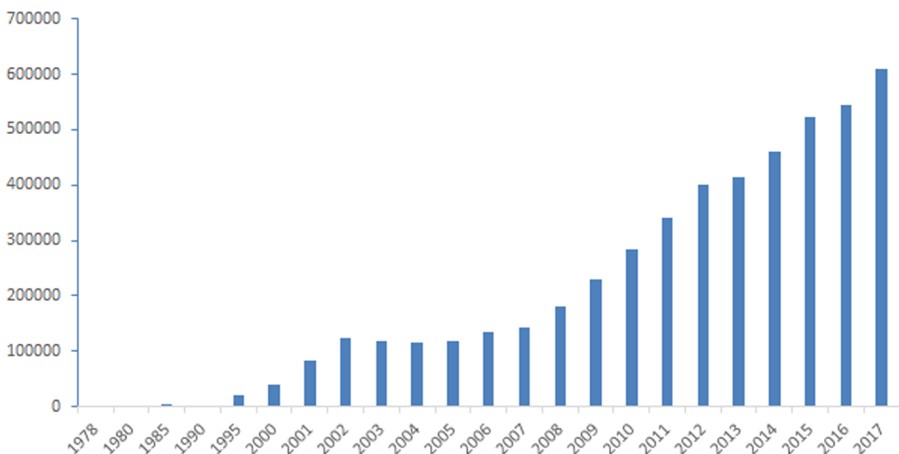

**Figure 1  Number of Chinese students studying abroad (1978–2017).**

Russian-English was better than that of English monolingual college students (*Kharkhurin, 2010*). This phenomenon is not confined to one country: in the United Arab Emirates (UAE), Farsi-English bilinguals revealed advantages in originality in thinking compared to native speakers from the same educational group (*Kharkhurin, 2009*). The Chinese scholar *Ni (2012)* combed through previous studies on the relationship between bilingual learning and creativity and suggested that students with a higher second language proficiency, earlier exposure to a second language and greater length of exposure to a second language have significantly improved creativity, such as in insightful problem solving (IPS).

## The influence of socioeconomic status (SES) on creativity

Prior studies have attempted to research creativity as a behavior resulting from an interaction among cognitive abilities, individuality, and social environment, but few studies have examined the role of SES, such as family income, in creative thinking (*Parsasirat et al., 2013*). Recently, some scholars have begun to explore the relationship between family income and creativity. For example, *Kim & Lee (2015)* reported that children from low-income families have fewer opportunities to engage in scientific experiences than other children; however, through creative science activity programs, their scientific attitude, self-esteem and self-efficacy will be significantly improved. SES is an important factor in the educational environment, and it is one of the crucial elements influencing creativity. SES can promote children's creative thinking as early as the age of 6 (*Jankowska & Karwowski, 2019*). *Castillo-Vergara et al. (2018)* investigated students from 75 educational institutions through Quality and Educational Context Questionnaires to determine SES; these questionnaires were mainly used to determine the educational level of the students' parents and the monthly income of their family. The researchers also used the Multifactorial Evaluation of Creativity (EMUC) test to measure the performance of creativity including fluency, flexibility, and originality. The results showed that as the socioeconomic level increases, so does the ability to innovate. Similarly, *Sánchez & Salinas (2008)* suggested that people with the highest SES levels perform better in the language arts, mathematics, and

science, while low-income groups show the opposite results. Based on the above research, we can speculate that good economic conditions in a family are a key factor in promoting the development of creativity, or that a good environment is at least a contributing factor. Undoubtedly, in mainland China, a stable income is necessary for studying abroad. Therefore, this study will investigate the income of the participants through questionnaires, and it will explore the relationship between income and creativity and the intention to study abroad.

## The present study

Since the reform and opening-up, China's economy has developed rapidly, and a booming economy has emerged. Meanwhile, with strong economic support, the government has paid increasing attention to cultivating high-quality innovative students. Thus, the state intends to improve students' creativity and academic performance through international exchange.

Bilingualism not only plays an important role in cross-language communication but also fulfills social culture and psychological cognitive functions. In recent decades, the relationship between bilingualism and cognitive ability has become a popular topic in pedagogy and psychology. Discussion of this issue has not reached a unanimous conclusion (*Paap, Johnson & Sawi, 2015*), but on the whole, most scholars believe that bilingual teaching and multicultural experiences do not burden learners or hinder their growth (e.g., *Alesina, Harnoss & Rapoport, 2016*; *De Bloom et al., 2014*; *Shen & Yuan, 2015*). In contrast, bilingual teaching and multicultural experiences are necessary cultural, economic, and political resources for both the country and the individual. Although the influence of bilingualism on creativity has been studied in depth, few studies have focused on the exploration of convergent creativity. Therefore, this study aimed to examine the impact of bilingual learning and willingness to study abroad on convergent creativity based on previous research.

## MATERIALS & METHODS

### Participants

In total, 76 healthy undergraduates (35 male; $age_M = 20.32$ years, $age_{SD} = 2.10$ years) participated in this study. Of the participants in this experiment who spoke Chinese as their first language, 66 (86.8%) participants spoke one foreign language (English), eight (10.5%) participants spoke two foreign languages, two (2.6%) participants spoke three foreign languages, and all participants spoke English as a second language. Eight of the 76 participants were excluded from the data analysis (10.53%) for the following reasons: One participant did not provide thoughtful responses to the RAT problems, and the other participants required longer than 20 s to answer one of the RAT problems (*Shen et al., 2016a*; *Shen et al., 2019*). A final sample of 68 (32 male; $age_M = 20.37$ years, $age_{SD} = 2.18$ years) participants was used in this study, including 28 students who planned to study abroad and 40 students who did not. All participants had normal or corrected-to-normal vision and had no experience with any similar experiments. Before the experiment, the participants provided written informed consent, and they received RMB 10 Yuan at the

end of the study. The study was approved by the Ethics Committee of Human Research Protection of Hohai University.

## Materials

### Convergent creativity task

According to Mednick's model, convergent creativity is primarily measured by the RAT; highly creative people show higher associative fluency, and they perform better on the RAT (*Benedek & Neubauer, 2013*; *Shen et al., 2016a*). A typical RAT contains three unrelated words (e.g., rat/blue/cottage), and participants are asked to find a fourth word (e.g., cheese) that serves as an associative link to the stimulus test (*Alici et al., 2019*; *Benedek & Neubauer, 2013*). In this study, 53 Chinese RAT problems (*Ding et al., 2014*; *Shen et al., 2016a*; *Shen et al., 2019*; *Shen et al., 2018*) were used as the experimental materials, including five problems (two difficult and three simple problems) used during the practice session. Forty-eight problems, including 24 difficult and 24 simple problems, were used in the formal experiment. These problems can occasionally lead to feelings of insight depending on the participants' judgment. Each item contained three different Chinese words (命/男/学, i.e., in English, *destiny/male/learning*) that prompted the participants to respond with a single solution character (生, i.e., in English, *person*). The complete solution consisted of three Chinese phrases (生命/男生/学生, i.e., in English *life/men/student*); participants can score one point for each correct answer (up to 48 points). When the score is higher, the performance of convergent creativity is better. The Cronbach's $\alpha$ of this test was 0.92, indicating strong internal reliability (*Shen et al., 2016b*). Furthermore, this test has been used in many studies (e.g., *Shen et al., 2016a*; *Shen et al., 2016b*), indicating it has good external validity. During this test, the provided problems (horizontally placed) and their corresponding solutions appeared in the center of the screen in boldface 24-point type, black on gray.

### Bilingual proficiency and overseas experience questionnaire

This questionnaire included 14 items regarding the basic information of the participants, such as their experiences with learning a foreign language, experiences with studying abroad, plans to study abroad, parents' overseas experiences, and monthly household income. The main purpose of this questionnaire was to collect basic information about the participants, especially regarding their intention to study abroad and second language proficiency. Each item required the participants to choose the most suitable option from the choices given. The item regarding second language proficiency, which had a total of five options ranging from very proficient (1) to not proficient (5), was emphasized. According to the Chinese English Scale (CES) (published in *The Results of Linking IELTS and Aptis to China's Standards of English Language Ability Press Conference*), the participants provided detailed explanations to the experimenter before the test. If the participant's International English Language Testing System (IELTS) score was greater than 7.5 or he or she had passed the College English Test-8 (CET-8) (mastering approximately 8,000 words), Option 1 was selected. If the participant had an IELTS score greater than 6.5 or they had passed the College English Test-7 (CET-7) and had mastered approximately 7,000 words, Option 2 was selected. If the participant passed the College English Test-6 (CET-6) or the College

English Test-4 (CET-4) (mastering approximately 5,000 words), Option 3 was selected. If the participant learned English in primary school but had not passed the CET-4, Option 4 was selected (mastering approximately 2,000 words). Finally, if the participant only knew a few English words (fewer than 500), Option 5 was selected (e.g., *Schachter, Kimbro & Gorman, 2012*; *Müller, 2016*). In special cases, if the participant found it difficult to judge their English level, it was assessed by a vocabulary test (*Hommel et al., 2011*). In this study, all participants were able to clearly determine their second language proficiency (all Chinese college students are required to take different English language tests). Thus, the participants did not take the vocabulary test.

### Positive and Negative Affect Scale (PANAS)

To exclude potential interference by pre-experimental emotions in the results, the Positive and Negative Affect Schedule (PANAS—State: *Watson, Clark & Tellegen, 1988*) was used. The PANAS contains 20 one-word adjective items reflecting positive affect (PA) and negative affect (NA), and each subscale includes 10 (positive/negative) items (*Watson, Clark & Tellegen, 1988*). The PA and NA subscales showed no correlation ($r = 0.14$), which corroborates the results reported in previous studies (e.g., *Heubeck & Wilkinson, 2019*; *Leue & Beauducel, 2011*). The respondents were asked to rate the items on a 5-point Likert-type scale (1 = very slightly or not at all to 5 = extremely). The alpha coefficients of the overall scale, the PA subscale, and the NA subscale were 0.86, 0.81, and 0.91, respectively, indicating strong internal consistency.

### Creative self-efficacy instrument

To rule out the possible impact of creative self-efficacy on the results of the RAT, this study used a four-item Creative Self-efficacy Instrument (*Tierney & Farmer, 2002*) measured on a scale ranging from 1 (strongly disagree) to 7 (strongly agree). A sample item was "Suggests new ways of performing work tasks". In a study conducted by Tierney and Farmer, this instrument showed a good level of reliability (manufacturing, $\alpha = 0.83$; operations, $\alpha = 0.87$). In this study, the Cronbach's $\alpha$ was 0.85.

## Procedures

The participants required approximately 30 min to finish the experiment. First, the participants completed a questionnaire about their basic information according to their actual situation. After this section, the participants were asked to complete the PANAS based on their current mood. Finally, the participants responded to the RAT problems, which were designed and presented by E-prime 2.0 (Psychology Software Tools, Inc., Pittsburgh, PA, USA).

Before the formal experiment, the participants completed a 5-item practice session to become familiar with the experimental process. Then, the participants completed 48 RAT problems as shown in Fig. 2. Each trial began with a " +" fixation in black on a computer screen to keep the participants focused on the screen. Then three words appeared in the center of the monitor in Song style, No. 28, black. The problem was displayed on the screen until the participants answered the problem or 20 s had passed. If the participant solved the problem, they could press the space bar on the keyboard and enter the answer stage

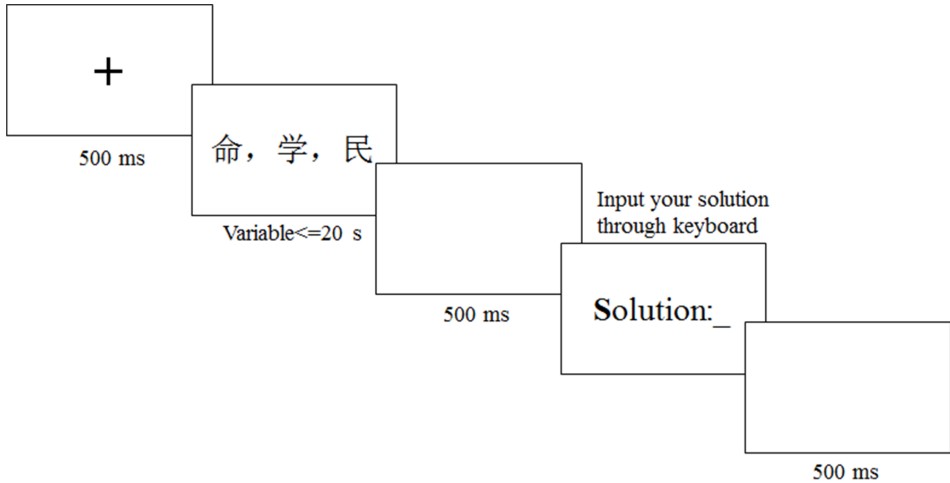

**Figure 2  Process of the convergent creativity task.**

after a 500 ms interval. On this screen, the participants had unlimited time to type in the answer, and they pressed "Enter" to indicate they had finished this stage. Finally, a 500 ms interval represented the end of the trial with a grayish screen.

## Statistical data analysis

Data were input and analyzed using SPSS Version 20.0 for Windows (SPSS Inc., Chicago, IL, USA). The difference between the group that planned to study abroad and the group that did not on RAT performance and language was analyzed through independent-samples $t$-tests. Second Language proficiency is a rank-ordered variable, and so between-group differences were tested by the Mann–Whitney $U$ test (*Isohashi et al., 2013*). The Pearson and Spearman correlation were used to analyze the relationships between the variables (*Puth, Neuhauser & Ruxton, 2014*). The mediating effect of second language proficiency was measured by nonparametric bootstrapping, and *Hayes*'s (*2013*) process was used in the analysis.

## RESULTS

### Correlations

The RAT is often used to explore the phenomenon of creative thinking, especially when examining insight into problem solving. The averages of the RAT score, monthly household income, and bilingual proficiency were 20.53 (SD = 5.90), 8842.65 RMB (SD = 10082.13), and 3.43 (SD = 0.70), respectively. Table 1 presents the correlations between the RAT performance and the individual differences measured in the study. One valuable result was that proficiency in a second foreign language was strongly correlated with the RAT score ($r = 0.45$, $p < 0.01$), as was the willingness to study abroad ($r = -0.29$, $p < 0.05$).
**Table 1** Correlations between RAT performance and individual variables.

| | RAT solved |
|---|---|
| Income | 0.13 |
| Second language proficiency | 0.45[**] |
| Willingness to study abroad | −0.29[*] |

Notes.
[*]$p < 0.05$.
[**]$p < 0.01$.
[***]$p < 0.001$.

**Table 2** The difference RAT performance and income in two groups.

| | Planning to study abroad ($n = 28$) | Not planning to study abroad ($n = 40$) | 95% CI | $t$ | Cohen's $d$ |
|---|---|---|---|---|---|
| | $M \pm SD$ | $M \pm SD$ | | | |
| RAT solved | $22.57 \pm 5.03$ | $19.10 \pm 6.10$ | $\pm 2.75$ | 2.48[*] | 0.62 |
| Income | $11767.86 \pm 14704.17$ | $6795.00 \pm 3807.75$ | $\pm 4756.95$ | 2.05[*] | 0.54 |

Notes.
[*]$p < 0.05$.

## Influence of willingness to study abroad on convergent creativity

This study examined the influence of overseas experiences[2] and willingness to study abroad on the participants' performance on the RAT. First, the results of the PA and NA subscales were analyzed to rule out the effect of natural emotions on the performance on the RAT. The group that planned to study abroad and the group that did not showed no difference on the PA ($t = -0.87$, $p = 0.39 > 0.05$, Cohen's $d = -0.21$) and NA ($t = 1.17$, $p = 0.25 > 0.05$, Cohen's $d = 0.31$) subscales. Thus, the additional variable of natural emotions was excluded from this study. The results of the Creative Self-Efficacy Instrument revealed no difference between the group that planned to study abroad and the group that did not ($t = 1.61$, $p = 0.11 > 0.05$, Cohen's $d = 0.40$).

Table 2 reports the difference in the performance on the RAT between two groups, a group of participants who planned to study or work abroad and a group of participants who did not. According to an independent-samples $t$-test, the participants who planned to study abroad had significantly higher scores than those who were not ($t = 2.48$, $p = 0.016 < 0.05$, Cohen's $d = 0.62$). This result suggests that people who want to study or work abroad perform better at solving RAT problems than those who do not. The average monthly household income of the participants who intended to study abroad was 11,767.86 yuan, while that of the participants who did not plan to study abroad was 6795.00 yuan, with a significant difference between the two groups ($t = 2.05$, $p = 0.044 < 0.05$, Cohen's $d = 0.54$). Thus, the family income of those who wanted to study abroad was significantly higher than that of the students who did not. Furthermore, as shown in Table 3, students who wanted to study abroad were significantly more proficient in English than students who did not ($p < 0.01$).

To further explore the influence of bilingual learning and willingness to study abroad on RAT problem solving, a backward regression analysis was employed to analyze the RAT problem scores (Tanksale, 2015). The monthly household income, number of foreign

[2]Overseas experiences were measured by the number of short-term (less than 2 weeks) and long-term (more than 2 weeks) studies or trips abroad by the participants (Shen & Yuan, 2015). In this study, only nine (13.3%) participants had short-term experience traveling abroad, and only two (2.9%) participants had been abroad for more than 2 weeks. Most participants did not have experience studying, traveling or working abroad.

**Table 3  The difference language proficiency in two groups (Mann–Whitney *U* test).**

| | 1 | 2 | 3 | 4 | 5 | *Mean rank* | *P*-value |
|---|---|---|---|---|---|---|---|
| Planning to study abroad ($n = 28$) | 1 | 2 | 17 | 8 | 0 | 27.41 | 0.006 |
| Not planning to study abroad ($n = 40$) | 0 | 1 | 15 | 22 | 2 | 39.46 | |

**Table 4  Results of hierarchical regression.**

| Model | Independent variable | Dependent variable | B | t | SE |
|---|---|---|---|---|---|
| 1 | Willingness to go abroad | Second language proficiency | 0.48 | 2.96[**] | 0.16 |
| 2 | Second language proficiency | RAT solved | −3.57 | −3.67[***] | 0.97 |
| | Willingness to go abroad | | −1.75 | −1.28 | 1.37 |

**Notes.**
[**]$p < 0.01$.
[***]$p < 0.001$.

languages spoken by the participants, willingness to study abroad, and number of times the participants and their parents had traveled abroad were input to predict the performance on the RAT problems. Only second language proficiency predicted the score on the RAT ($B = 0.47$, $SE = 0.92$, $\beta = 0.47$, $t = 4.36$, $p < 0.001$).

## Mediating effect of second language proficiency

According to *Hayes (2013)*, this study adopted nonparametric Bootstrap to explore the mediating effect of second language proficiency between willingness to go abroad and RAT solved, which was executed based on the PROCESS macro and the observed variables (*Jach et al., 2018*). By sampling the original data, 5,000 samples were extracted to estimate the 95% confidence of the mediation effect test (*Calvo-Mora et al., 2014*; *Yang, Liu & Chen, 2018*). The 95% confidence (LLCI = −3.30, ULCI = −0.61) of the indirect effect did not include 0; thus, the mediating effect of second language proficiency was significant. The regression analysis results are shown in Table 4. According to regression model 1, willingness to go abroad has a significant predictive effect on second language proficiency. According to model 2, when both willingness to go abroad and second language proficiency are included in the model, the predictive effect of the RAT solved was significant, but the direct effect of willingness to go abroad on the RAT score was not significant; therefore, second language proficiency had a complete mediating effect between the two variables. The mediation analysis results of each outcome variable are displayed in Fig. 3 through a presentation of the unstandardized path coefficients of each model.

## DISCUSSION

The purpose of this study was to experimentally investigate the relationships between bilingual learning, willingness to study abroad and performance on a Chinese variant of the RAT problems as an index of convergent creativity. This research showed that bilingual learning was positively correlated with RAT performance. Students who were willing to study abroad performed better on the RAT than students who were not, but this effect was mediated by second language proficiency. On the basis of previous studies, the results are
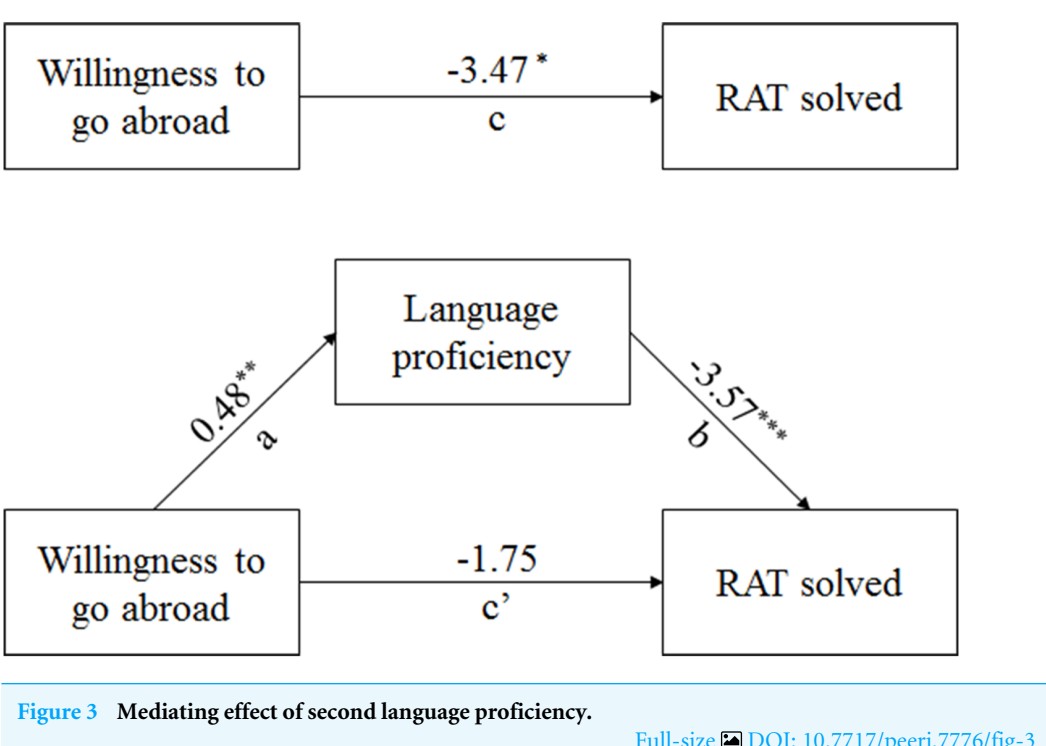

**Figure 3   Mediating effect of second language proficiency.**

discussed according to the aspects of bilingual learning, overseas experience and family income.

## The relationship between bilingual learning and convergent creativity

Bilingual learning is a requirement for compulsory education in China. As early as 1979, *Cummins (1979)* proposed the advantages of bilingual learning, which can help learners form two sets of language systems in their brain to create a special cognitive mechanism. In the process of language selection, the suppression of the nontarget language can improve the suppression control function of bilingual learners. Similarly, practicing a target language can strengthen the function of selective focusing. Moreover, code-switching between two languages helps individuals overcome psychological patterns and achieve strategy adjustment and conversion (*Costa, Hernández & Sebastián-Gallés, 2008*). Many studies have explored the impact of multilingual learning on creative thinking and its mechanisms of action. *Ricciardelli (1992a)* performed a meta-analysis of 25 studies published between 1965 and 1992 and found that approximately 80% of bilingual and multilingual learning experiences can improve individuals' creative thinking. Moreover, this conclusion was confirmed in the author's experiment (*Ricciardelli, 1992b*), which found that highly proficient bilinguals scored significantly higher on fluency in creative thinking, imagination, and language fluency than monolingual or low-proficiency bilinguals.

This study provided an important finding about convergent creativity in the literature: Second language proficiency is strongly correlated with the RAT score and can predict performance on RAT problems. On the one hand, this result is consistent with some previous findings, such as those reported by *Hommel et al. (2011)*. The authors'

study included 42 young healthy participants who were asked to complete the RAT and an English vocabulary test. The results showed that high-proficiency bilinguals performed better than low-proficiency bilinguals on the RAT. A study involving Russians (*Spanakos, 2001*), in which 278 middle school students were recruited as participants and a series of tests was used to measure their divergent and convergent creativity, obtained similar results. These results showed that bilingual participants outscored monolingual participants on convergent creativity. On the other hand, some studies, such as that conducted by *Vries & Lubart (2019)* to investigate scientific creativity, have reached the opposite conclusion. Interestingly, these authors found that culture-related variables were significantly negatively correlated with the originality of divergent and convergent scientific creativity and noted that the reason may be the mediating role of personality, personal information, and cultural adaptation. Finally, regarding the prediction of second language proficiency, *Wang & Cheng (2016)* drew a conclusion similar to that found in this study: English proficiency can significantly predict metaphoric creativity. This finding suggested that English ability is an important factor in predicting creativity. In summary, most studies have reported the influence of multilingual or linguistic learning on creative thinking, especially convergent creativity (*Shen & Yuan, 2015*).

## The relationship between studying abroad and convergent creativity

Numerous studies have confirmed that overseas experiences can influence people's creative thinking. For example, people with one year of immigration experience had a significantly higher score on creative thinking fluency than those without such experience (*Fee & Gray, 2012*). Surprisingly, in an examination of the impact of short-term overseas experience on individual creative thinking with a traceable pre-post-test design, *De Bloom et al. (2014)* found that studying abroad for only two weeks can promote fluency. Regarding convergent creativity, *Maddux & Galinsky (2009)* explored the effect of foreign experiences on the RAT. The results showed that a sample of individuals who had lived abroad performed significantly better on the RAT than a sample of people who had not lived abroad. In addition, this temporary facilitative effect of foreign experiences was strongest among the participants who had lived abroad the longest. In summary, both long- and short-term overseas experiences may have a positive effect on individual creative thinking.

The above results could not be analyzed in this study because the proportion of participants with overseas experience was too small: only 13.2% have short-term overseas experience and 2.9% have long-term overseas experience. All participants were students (56 undergraduates and 10 postgraduates). Although the participants were adults, they often had insufficient social experience, limited living expenses and heavy academic pressure; thus, the participants may not have had the opportunity to study abroad. Therefore, most participants did not have experience studying, visiting, or working abroad. However, as expected, cultural factors were related to convergent creativity; people who want to go abroad tend to have stronger critical thinking skills and a stronger desire for expressing themselves, such as being adventurous, culturally savvy or globally cosmopolitan (*Adam et al., 2018*; *Martin, Katz-Buonincontro & Livert, 2015*). *Frändberg (2015)* conducted biographical narrative interviews with 22 adults; they found that the

decision regarding the plan to study or work abroad was typically described as highly personal. People with this intention usually believe they can improve themselves and develop creative ideas in a foreign country. Notably, the intention to study abroad was significantly positive correlated with the performance on the RAT. This finding corresponds to the study of *Lee, Therriault & Linderholm (2012)*, who used the Abbreviated Torrance Test for Adults (ATTA) and the Cultural Creativity Task (CCT) to measure the creativity level of three types of participants: those who have studied abroad, those who plan to study abroad, and those with no plan to study abroad. On both scales, the results showed that students who plan to study abroad scored higher than students who did not.

The reason for the above finding might be related to the mediating effect of second language proficiency. To face the challenges of an increasingly globalized world, second language proficiency is becoming increasingly important for students who want to obtain better educational opportunities (*Schoepp, 2018*). In China, to study abroad, students must pass an English test, such as the IELTS or the Test of English as a Foreign Language, to prove that they have the language ability to live and study abroad. Through interviews with 32 students in Guangzhou who want to go abroad, *Wu (2018)* found that this group wanted to improve their second language proficiency through overseas study experience, and they were willing to invest time into passing the English tests. This finding may be because excellent performance on the English tests (e.g., IELTS) not only demonstrates students' English ability but also predicts their academic success (*Schoepp, 2018*). Therefore, students must exert considerable effort to improve their English skills so they can perform well on English tests.

Going abroad is an opportunity for individuals to come into contact with a strange environment, characterized by novelty and accompanied by completely different values, cultural identities and behavioral habits (*Adam et al., 2018*). Individuals who want to go abroad are usually more open to new experiences; for example, they are more intellectually curious about foreign cultures and more receptive to cultural instruction, increasing the irregularity and cultural relevance of their approach to insight problems (*Cho & Morris, 2015*; *Martin, Katz-Buonincontro & Livert, 2015*). *Silvia et al. (2009)* suggested that openness to new experiences plays an important role in the performance of creativity, with two main aspects: openness (including imagination, creativity, and aesthetics) and intellect (including thinking and reasoning). Their experiment included 189 students (*Silvia et al., 2009*), and their personalities, divergent thinking, creative achievement and creative self-efficacy were measured through questionnaires. From the latent variable models, they found broad effects of openness to experience to creativity. Similarly, the results from *Schilpzand, Herold & Shalley*'s (*2011*) study with 31 graduate student teams indicated openness is significantly related to team creativity. For intellect, individuals can improve their cultural intelligence by mastering and transmitting their multicultural experience (*Hu et al., 2017*). Cultural intelligence makes individuals aware of cultural differences, and they can master other cultural knowledge well, which provides them with new ways of thinking and with new concepts. These new inputs create more possibilities for them to view things from different perspectives and contribute to the performance of creativity (*Cheng & Leung, 2013*; *Hu et al., 2017*). Through a questionnaire survey of 310 international students, *Hu*

*et al. (2017)* found that cultural intelligence was significantly correlated with creativity, partially mediating the relationship between multicultural experiences and creativity. Based on the above studies, people who intend to go abroad may perform better on the RAT because of their larger openness to experience.

Family income is a key factor that should be considered in study abroad plans and even in the preparation of bi- or multilingual learning. Family income performs an important function in the educational and growth processes. For example, in China, high-income families receive the greatest educational benefits, while low-income families have the lowest investment in children's specialty training (*Zhang et al., 2015*). Similarly, family income has a significant independent predictive effect on early childhood language ability after controlling for children's age and gender (*Li, Li & Li, 2012*). Income is also related to children's cognitive development in foreign countries, as shown by the Panel Study of Income Dynamics (PSID) in America. The study showed that the level of income was associated with Woodcock-Johnson (W-J) Achievement Test scores, and that income stability was associated with W-J applied problem scores and the Behavior Problem Index (BPI), even after including all controls in the models (*Yeung, Linver & Brooks-Gunn, 2002*). In addition, among the different indicators of SES, only income showed significant associations with children's emotional health status. Therefore, good and stable economic conditions are crucial for studying abroad (*Hercog & Van de Laar, 2017*) and can provide material security for students to study abroad and achieve professional and personal goals (*Doppen, An & Diki, 2016*).

Over the last 10 years, the number of Chinese students studying abroad has grown rapidly, but more than 90% of these students choose to study abroad at their own expense. This number is mainly determined by the economic strength of their families. Therefore, family income is the most important factor affecting the number of students studying abroad (*Li, 2018*). In China, among children under the age of 15, the proportion of children who travel abroad for further study is 13.02%, and the proportion of children who may be sent abroad in the future is 23.58%. Families with such plans are often high-income families (*Sun et al., 2016*). The influence of high family income on creativity might be another reason for the significant positive correlation between the willingness to study abroad and convergent creativity.

## CONCLUSIONS AND FUTURE RECOMMENDATION

In conclusion, this study reveals that bilingual learning is positively correlated with convergent creativity. When a student's English level is higher, the student scores better on convergent creativity as measured by the RAT. Students' plans to study abroad and their convergent creativity abilities are significantly related, with students who want to study abroad performing better on the RAT. This facilitation might relate to the mediation of second language proficiency.

At present, few studies have examined the relationship between bilingualism and convergent creativity. This study not only explored this issue but also searched for a connection between convergent creativity and life plans.

This study had several limitations. First, in this study, we used the PANAS and Creative Self-Efficacy Instrument to control for the influence of emotions and creative self-efficacy on the study. Emotions are an important factor affecting individuals' creativity (e.g., *Lin et al., 2014*), and the PANAS is a common tool used to measure emotions in many studies (e.g., *Ceci & Kumar, 2016*; *Fernandez-Abascal & Diaz, 2013*). Creative self-efficacy has a positive and direct influence on achievement goals (*Bang & Reio, 2017*; *Puente-Diaz & Cavazos-Arroyo, 2018*) and other personality factors (*Karwowski et al., 2013*). In many studies investigating creativity, creative self-efficacy is used as a moderator variable (e.g., *Gong, Huang & Farh, 2009*; *Wang, Tsai & Tsai, 2014*). However, measuring more irrelevant variables, such as novelty-seeking and the motivation to study abroad, is necessary to further ensure the accuracy of the results and to clarify the underlying mechanisms leading to the results. Second, in China, all college students are required to take different English language tests; thus, assessing second language proficiency using the Bilingual Proficiency and Overseas Experience Questionnaire is effective and convenient. However, if other groups of participants are included, such as workers or children, it is best to use a vocabulary test (*Hommel et al., 2011*). Third, this study showed that the intention to study abroad was related to performance in convergent creativity; in future research, the causes of this result could be explored by measuring other relevant variables, such as openness to experience. Furthermore, a considerable number of studies from diverse groups has validated the effectiveness of such a tool to measure the convergent creativity of Chinese people (*Ding et al., 2014*; *Shen et al., 2016a*; *Shen et al., 2018*; *Shen et al., 2019*; *Wu, Chang & Chen, 2017*). With the deepening of research on creativity worldwide, an increasing number of people may be interested in the creativity of Chinese people, so this tool must be further promoted and verified. To contribute to the attenuation of the controversy on this hot issue (*Paap & Greenberg, 2013*), further studies are also needed to explore the cognitive and neural mechanisms of bilingual learning and the willingness to study abroad on convergent creativity and the difference between convergent creativity and divergent thinking. This study explores only the relationship between variables. Thus, causal inferences are difficult to make. The reasons for the correlations between these variables may be possible to identify through cognitive and neural mechanisms. Through a more comprehensive analysis of these issues, effective measures to enhance students' creativity may be proposed from the perspective of educational policy.

### Funding

This work was supported by the Ministry of Education Humanities Social Sciences (18YJA190008), the National Natural Science Foundation of China (31500870), the Natural Science Foundation of Jiangsu Province (BK20181029), the Fundamental Research Funds for the Central Universities (2017B14514), China Postdoctoral Science Foundation (2017M621603), China Scholarship Council Foundation (201706715037), and Natural Science Foundation of Jiangsu College of China (17KJB190002). The funders had no role

in study design, data collection and analysis, decision to publish, or preparation of the manuscript.

## Grant Disclosures

The following grant information was disclosed by the authors:

Ministry of Education Humanities Social Sciences: 18YJA190008.

National Natural Science Foundation of China: 31500870.

Natural Science Foundation of Jiangsu Province: BK20181029.

Fundamental Research Funds for the Central Universities: 2017B14514.

China Postdoctoral Science Foundation: 2017M621603.

China Scholarship Council Foundation: 201706715037.

Natural Science Foundation of Jiangsu College of China: 17KJB190002.

## Competing Interests

The authors declare there are no competing interests.

## Author Contributions

- Yuan Zhao conceived and designed the experiments, performed the experiments, analyzed the data, contributed reagents/materials/analysis tools, prepared figures and/or tables, authored or reviewed drafts of the paper, approved the final draft.
- Yuan Yuan conceived and designed the experiments, analyzed the data, contributed reagents/materials/analysis tools, prepared figures and/or tables, authored or reviewed drafts of the paper, approved the final draft.
- Wangbing Shen conceived and designed the experiments, performed the experiments, analyzed the data, contributed reagents/materials/analysis tools, prepared figures and/or tables, authored or reviewed drafts of the paper, approved the final draft, the cost of the subjects was provided.
- Chuanlin Zhu contributed reagents/materials/analysis tools, prepared figures and/or tables, approved the final draft.
- Dianzhi Liu analyzed the data, contributed reagents/materials/analysis tools, prepared figures and/or tables, authored or reviewed drafts of the paper, approved the final draft, the cost of the subjects was provided.

## Human Ethics

The following information was supplied relating to ethical approvals (i.e., approving body and any reference numbers):

The Ethics Committee of Human Research Protection of Hohai University granted approval to carry out the study within its facilities.

## Data Availability

The raw measurements are available in the Supplemental Files.

## Supplemental Information

Supplemental information for this article can be found online at http://dx.doi.org/10.7717/peerj.7776#supplemental-information.

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
