# Peer review of "The relationships between bilingual learning, willingness to study abroad and convergent creativity"

_PeerJ, doi:10.7717/peerj.7776_

## Round 0.1 · original submission · Minor Revisions

Thank you for your submission to PeerJ. I have reviewed the manuscript and received reviews from two expert reviewers, who are generally in agreement that the research is novel and warrants publication, but needs some work. Both reviewers did a careful job and identified numerous small issues, and these all seem reasonable.

My major concern is that although this work is fundamentally correlational, throughout the manuscript causal attribution is made. A mediation model will not untangle causality. For example, table 3 caption "influence of willingness to study abroad on second language proficiency" implies that willingness to study abroad is impacting second language proficiency--it could easily be reversed.

Similarly, line 457: "plans to study abroad can also affect their convergent creativity ability". Alternately, convergent creativity may affect plans to study abroad. The entire section (line 359) is likewise misnamed. A reasonable hypothesis is that income increases language ability, increases willingness to study abroad, and increases RAT scores indirectly through many different educational activities.

·

Basic reporting

It is a very interesting article, which addresses a topic of interest such as creativity. However, there are some suggestions that should be addressed before the article is published.

In the introduction, the authors pose the case of Rohan as an example of experience to sustain the importance of cultural activities. The case is interesting, but as stated, it is not possible to establish scientific sustenance.

In the introduction the authors suggest that a stable income is necessary to study abroad, however, the design of the experiment does not clearly control this variable. The inference of the concept is not clear.

They also infer that people study more and more abroad or plan to study abroad for further studies, but this inference lacks scientific support in the article.

Experimental design

As an objective it is indicated that the study seeks to measure the impact of bilingual learning and the willingness to study abroad on convergent creativity. However, I believe that the results of the objective have not been explained clearly. The analyzes show significant differences between groups and correlations, but I have not been able to visualize the impact indicated clearly. The discussion starts with: This research showed that bilingual learning was positively correlated with convergent creativity performance, that is, correlation and not impact. Even when a procedure of mediating effect is established, there are doubts about the application of the methodology.

The use of Bootstrap of 5,000 samples is proposed to measure the effect. However, the method, the technique used, the software, are not presented. It seems to be a PLS analysis, with which the model must report what is indicated in the literature regarding the validity of measurements and the model. There is a lack of information to clarify the technique.

It is not indicated in the procedure who and what the Bilingual Proficiency and Overseas Experience Questionnaire was used for. It must be explained in more detail.

Validity of the findings

The Cronbach's alphas of the PA and NA subscales were 0.28 is a value that must be explained.

When the technique used for mediation is indicated, the analyzes must be reported to validate said results.

Additional comments

In a very interesting article. But I feel that there is a lack of background in its presentation for its final publication.

Reviewer 2 ·

Basic reporting

The manuscript studied the impact of willingness to study abroad and English proficiency on students’ convergent thinking performance. Moreover, the relationship between willingness to study abroad and RAT performance was found mediated by second language proficiency.
Overall, the study investigated the relationship between bilingualism and convergent creativity from a novel perspective that could be of general interest in the field of pedagogy and psychology. The results are interesting and worth to be published.

Experimental design

While the study design is quite simple and straightforward, more explanations on the following issues are necessary:
1. In the introduction, the author provided evidence for how multicultural experiences can make a difference in individual creative ability. However, it should be noted that there is a gap between the experiences and willingness to studying abroad. Therefore, the fact that multicultural experiences improved creativity is not sufficient to support the hypothesis that willingness to study abroad can improve creativity. Although the author suggested that the difference in willingness might reflect a difference in openness, the reason for involving the willingness to study abroad in the current study should be discussed more explicitly with more empirical evidence.
2. Please provide a descriptive result about RAT performance, monthly household income, and bilingual proficiency.
3. In the method section, information about the measurement of monthly household income is needed.
4. It should be “PA” instead of “PN” in line 282.
5. The importance of SES was discussed in the introduction, however only the mediating effect of second language proficiency was analyzed. Why not take the mediating effect of monthly household income into consideration?

Validity of the findings

6. In the conclusions section, it was mentioned that “education policies should focus on cultivating students’ English proficiency and increasing opportunities for students to study and work abroad because these are effective ways to improve students’ convergent creativity”. Considering the quasi-experimental design of the current study, more experiments should be done before drawing this conclusion.

---

## Round 0.2 · Minor Revisions

Two reviewers have examined the revised manuscript and agree that it is improved. One has suggested some revisions to help organize the introduction. I have reviewed the manuscript carefully and agree with this; many of the important issues are not discussed until the discussion, and these issues are made clear by that organization.

I have also carefully reviewed the manuscript and attach my track-changes version below. I do not think there are substantial problems with the data, but I remain skeptical about:
1. whether RAT measures creativity
2. whether a correlation between RAT and bilingualism or study abroad (either desire or actual study) implies a causal relationship; there are still places in the manuscript that imply this.
3. Whether the chinese-translated RAT is fair, and does not naturally advantage english-language speakers to perform better.


One of the reasons for my skepticism is that there has been recent criticism of similar results within research on the putative benefits of bilingualism on executive control. This work is not cited, although there are points where it should be.I suspect that many of the same methodological issues exist in this area. See Paap and other references (I think there was another review paper or meta-analysis within the last year).

Paap, K. R., & Greenberg, Z. I. (2013). There is no coherent evidence for a bilingual advantage in executive processing. Cognitive psychology, 66(2), 232-258.

Paap, K. R., Johnson, H. A. & Sawi, O. Bilingual advantages in executive functioning either do not exist or are restricted to very specific and undetermined circumstances. Cortex 69, 265–278 (2015).


* The very strong correlation between second-language proficiency and RAT performance deserves discussion. The problem is that there is little evidence that it has much to do with creativity. RAT is a language/memory search and retrieval task, which is often used as to induce insight state in problem solving. Because of this, the original English RAT is sometimes used as a between-subjects measure of creativity, but what about the Chinese RAT? I think that the question is NOT answered with this study. Bilingualism confers an advantage for solving Chinese RAT problems. I don’t know the problems well enough to determine why, but quite possibly, there are aspects of English that are slipping in to the Chinese test, and so English fluency allows subjects to solve those problems. The way to test this is to use a group of chinese bilinguals who do not speak English. There are certainly such individuals, but they would also likely differ along other dimensions.
Another possibility (not mutually exclusive) is that English language proficiency is either associated with Chinese language skill; or another relevant skill like activating memory associations or memory retrieval. The causality could go in either direction; those who excel at learning english excel at language in general, or those who study English a lot gain cognitive skills that are strengthened by virtue of learning this second language. Most of those skills are not about creativity, but many of them could confer an advantage for the Chinese RAT problems. A similar story exists for the relationship between the Chinese RAT and interest in studying abroad.

Finally, the 'limitations' of the study in the discussion needs to address these issues.

Other issues:

Analyses:
* Please provide a key for the data files. I had a very difficult time checking the analysis because the values were not clear. The two different data files are not easily alignable because of participant drop-out.

* The pearson correlations should really be spearman; one variable is rank-ordered; another (income) has a very large outlier. I’d prefer a figure showing the scatterplots over just the correlation.

* The reference section is poorly formatted. I have added many edits to improve this, but do not think I found everything.


I have attached a marked-up pdf of the word document. It has marked changes and comments within the document. The major issues are incorporated in this letter, but some of the comments in the pdf are simply commentary that may not need to be addressed. You can feel free to address the issues discussed here in your response letter. If you'd like the word document so you can more easily incorporate the changes, email me at shanem@mtu.edu--the peerj interface does not allow me to upload a word document.

·

Basic reporting

I have reviewed again and I think there are some minor adjustments that the authors must incorporate before the article is published.

These are detailed below:
The introduction has been strengthened, but now I miss incorporating a section in which each of the hypotheses that we want to support is discussed. It is necessary to order the introduction and relevance of investigating the topic and the existing gap. But they are confused with the hypotheses. In the discussion these sections are very clear, I recommend doing this in the introduction. This will strengthen your work.
Line 110. Most studies are indicated, however, only one study is referenced. They must improve that phrase.

Experimental design

no comment

Validity of the findings

no comment

Additional comments

The document has improved greatly since its last presentation. I thank the authors for the effort to improve their work and take into account the observations.

Reviewer 2 ·

Basic reporting

no comment

Experimental design

no comment

Validity of the findings

no comment

Additional comments

All my concerns properly addressed, no further comment. Acceptance is suggested.

---

## Round 0.3 · accepted · Accept

One note: in the abstract, 'maybe' should be 'may be':

These findings suggest that improving students’ English proficiency and increasing their opportunities to study abroad may be effective ways to promoting convergent creativity.